# Prevalence of Respiratory Infections during the 2018–2020 Period in the Paediatric Population of Primary Care Centres in Central Catalonia

**DOI:** 10.3390/healthcare11091252

**Published:** 2023-04-27

**Authors:** María José Macías Reyes, Josep Vidal-Alaball, Eduardo Alejandro Suwezda, Queralt Miró Catalina, Maria Homs, Anna Ruiz-Comellas

**Affiliations:** 1University Hospital of Igualada, 08700 Igualada, Spain; 2Faculty of Medicine, Vic-Central University of Catalonia, 08500 Vic, Spain; 3Health Promotion in Rural Areas Research Group, Institut Català de la Salut, 08272 Sant Fruitós de Bages, Spain; 4Central Catalonia Research Support Unit, Jordi Gol i Gurina University Institute for Research in Primary Health Care Foundation, 08007 Barcelona, Spain; 5Primary Care Centre Vilanova del Camí, 08788 Vilanova del Camí, Spain; 6Primary Care Centre Sant Joan de Vilatorrada, 08250 Sant Joan de Vilatorrada, Spain

**Keywords:** respiratory infections, SARS-CoV-2, COVID-19, primary care

## Abstract

Following the COVID-19 pandemic, policies such as social distancing, hand washing, and the use of masks were implemented, which could play an important role in the reduction of infectious diseases. An observational, descriptive, cross-sectional study was conducted to observe the prevalence of respiratory infections in children under 15 years of age during the 2018–2020 period in Primary Care centres in Central Catalonia. In 2020, there was a 44.3% decrease in total consultations for respiratory infections compared to 2019. All respiratory infections exhibited a significant decrease except flu-like syndrome; children between the ages of 6 and 12 had the highest prevalence of flu-like syndrome (87.6%), and the SARS-CoV-2-19 infection was most frequent (4%) among those between the ages of 12 and 15. Compared to urban centres, rural centres presented a higher prevalence of all infections except flu-like syndrome and SARS-CoV-2. In conclusion, the COVID-19 pandemic caused a significant decrease in the number of consultations for respiratory infections in the paediatric population, except for flu-like syndrome, which increased in cases in January, February, and March 2020. No differences were found between sexes, although differences were found in the distribution of the different age groups.

## 1. Introduction

Respiratory tract infections are one of the main causes of morbidity in the paediatric age group and account for a large part of the healthcare activity in primary care and hospitals [1]. They occur mainly in the form of winter epidemics, caused by low temperatures that facilitate the increase in pathogens. Transmission occurs via respiratory droplets or by direct contact with contaminated objects [2].

Most respiratory infections are of viral aetiology and produce a similar and non-specific symptomatology. It is therefore often difficult to distinguish by clinical criteria alone what the cause is [3]. This leads to the use of multiple diagnostic tests and often to the prescription of unnecessary antibiotic treatments [4], generating costs in terms of healthcare and economic resources [5]. The impact of these infections on public health is difficult to calculate, causing an increase in first visits, repeated check-ups, school and work absenteeism, and, in addition, multiple prescriptions of antibiotics, antitussives, antipyretics, mucolytics, nasal decongestants and antihistamines [6].

In December 2019, a new coronavirus, severe acute respiratory syndrome coronavirus type 2 (SARS-CoV-2), which causes the disease known as COVID-19, was detected in China. The first case of SARS-CoV-2 infection was confirmed in Spain on the 31st of January 2020 [7]. On 11 March 2020, it was declared a pandemic by the World Health Organization (WHO) [8]. This virus is transmitted primarily through respiratory droplets and close contact [9]. In order to limit its spread, policies such as social distancing, hand washing, the use of masks and even measures such as school closures, home isolation, or limitation of extracurricular activities, among others, were implemented [10]. Mask use is effective in preventing respiratory infections by minimizing the excretion of respiratory droplets [11]. The masks act by preventing the spread of viral particles from asymptomatic patients to others, and reduce the viral inoculum, in such a way that the exposed person will develop a milder disease [12].

These restrictive measures may have played an important role in the decline of other childhood infectious diseases during the initial phases of the pandemic, producing a significant reduction in the total number of visits [13,14]. This decline in the number of consultations may also have been influenced by changes in parental care-seeking criteria and, in addition, by the fall in the overall burden of infectious diseases in childhood [15]. Liguoro et al. [13] noted that the total number of visits in the hospital emergency department of Udine (Italy) decreased by 73% from 2019 to 2020. Similarly, Lopez et al. [16] described a 47% reduction in total visits in primary care centres in Central Catalonia (Spain) during that period. Hibiya et al. [15] demonstrate that public health interventions have had a potential positive impact on controlling the COVID-19 pandemic by preventing transmission of droplet-borne infections.

We have not found studies describing the impact of COVID-19 on respiratory infections in the paediatric population in primary care in Spain. The main objectives of this study are to estimate the prevalence of respiratory infections between 2018 and 2020, and to compare the distinctions between the different infections in the first year of the pandemic to assess how COVID-19 has affected them. Other secondary objectives are as follows: to calculate the percentage change in prevalence in 2020 with respect to previous years, to assess the differences in the distribution of respiratory infections among different age groups (0–1, 1–6, 6–12 and 12–15 years old), and to observe the difference in prevalence of infections in rural versus urban primary care centres.

## 2. Materials and Methods

An observational, descriptive, cross-sectional study was conducted during the 2018–2020 period in the primary care centres of the Catalan Institute of Health of Central Catalonia, Spain. The reference population was 523,328 inhabitants divided among the regions of Anoia, Moianès, Osona, Berguedà, and Bages, of which 80,884 were under 15 years of age.

The study included patients under 15 years of age who consulted the Primary Care centres of Central Catalonia for respiratory clinical conditions during the period between 1 January 2018 and 31 December 2020, and who recorded in their clinical history the diagnosis coded according to ICD-10 of: common cold (J00), sinusitis (J01), acute pharyngitis (J02), tonsillitis (J03), laryngotracheitis (J04), influenza (J09-11), pneumonia (J12-18), acute bronchitis (J20), bronchiolitis (J21), otitis (H65-66), stomatitis (B08), or COVID-19 (V07).

To calculate the prevalence, the total cases of each pathology among the total population under 15 years of age assigned to Central Catalonia that year were considered. Prevalence was estimated for the total population and was segmented according to the main sociodemographic variables. Prevalences between 2018 and 2019 and between 2019 and 2020 were compared by means of a proportion test. The percentage change has been calculated as (final value − initial value)/initial value.

The confidence intervals of the study were 95% and the significance level was 5%. All analyses were performed with version 4.2.1 of the statistical program R.

Other sociodemographic variables studied were age, sex, consultation date and type of primary care centre (rural or urban). Rural areas were defined as areas with less than 10,000 inhabitants and a population density of less than 150 inhabitants/km^2^ [17].

Data collection and information sources:

The variables necessary to carry out the study were obtained from the Primary Care information systems of the Health Region of Central Catalonia belonging to the Catalan Health Institute, after a process of pseudonymisation by the institution’s information systems. Ethical approval was obtained from the IDIAP Jordi Gol committee, code 21/041-P.

## 3. Results

Between 2018 and 2020, there were 110,478 visits for respiratory infections in children under 15 years of age, all of which were included in the study. A decrease was observed in the total number of consultations for respiratory infections in the paediatric population, with a total of 44,229 in 2018 and 42,563 in 2019, while in 2020, they totalled 23,686, which was a decrease of 44.3% compared to 2019 and 46.4% compared to 2018.

When analysing the distribution of all respiratory infections by month (Figure 1), we observed a general decrease in the number of consultations. In 2018 and 2019, there was a decrease in summer, especially in the months of July and August. In 2020 there was a clear decrease, more marked than the previous ones, from March onwards, which coincides with the beginning of the SARS-CoV-2 pandemic and the lockdown in Catalonia.

All respiratory infections, except flu-like syndrome, showed similar prevalences in the population studied in 2018 and 2019 and a marked decrease in 2020. Table 1 shows the prevalence of the respiratory infections analysed. In the case of flu-like syndrome, in 2020, there was a prevalence of 3.8 (CI: 3.73;4.03), which was higher than in 2018 with 2.8 (CI: 2.7;2.96) and in 2019 with 2.7 (CI: 2.6/2.86).

In 2020, compared to 2019, a statistically significant reduction was observed for all respiratory infections except flu-like syndrome, which suffered an increase of 41.7% (*p* < 0.001). The most marked variation in 2020, compared to 2019, can be observed in stomatitis with a decrease of 80.6% (<0.001), acute tonsillitis and streptococcal pharyngitis, with a decrease of 60.4% and 69.9%, respectively (*p* < 0.001).

In the separate analysis by gender, no differences were observed, either in terms of the distribution of diseases or in the percentage variation in prevalence (Appendix A) in 2020, compared to 2019.

By age group (Table 2 and Table 3), the variation in the prevalence of infectious pathologies analysed in all age groups can be observed, with an unequal distribution in the frequency of infections in the different age groups, but no correlation was observed between the incidence of infections and the age of the children analysed.

In children under one year of age, bronchiolitis predominates with a prevalence of 21.2 and 26.2 in 2018 and 2019, respectively, decreasing to 9.1% in 2020. Children in their first year of life constitute the only subgroup in which the prevalence of flu-like syndrome did not increase in 2020 with 1.7, similar to 2018, and lower than in 2019 with 1.8. The prevalence of COVID-19 was 3.3 (CI: 2.67;4.16) (Table 2). For the population between the ages of 1 and 6, the prevalence of SARS-CoV-2 was 2.5 (CI: 2.33;2.79). This is the age bracket with the highest prevalence of acute tonsillitis, with 3.1 in 2020, and 8.9 and 7.3 in 2018 and 2019, respectively (Table 2). The prevalence of flu-like syndrome was highest in patients between the ages of 6 and 12 in 2020, with 3.8, compared to 2.2 and 2 in previous years. With respect to COVID-19, the prevalence was 3.3 (CI: 3.16;3.6). The group of adolescents between the ages of 12 and 15 had the highest prevalence of COVID-19, with 4 (CI: 3.79;4.35) (Table 3).

When evaluating the variation in the prevalence between 2020 and the previous year (Table 1), we can observe that, in children under one year of age, the biggest decrease in diagnoses is that of acute bronchiolitis, of 65.1% (<0.001). On the contrary, no changes were observed in streptococcal pharyngitis, flu-like syndrome, or pneumonia (Table 2). In pre-schoolers between the ages of 1 and 6, the most marked decreases were in stomatitis, with 82.6% (*p* < 0.001), and in acute sinusitis, with 75% (*p* < 0.001). In this group, we can observe the biggest reduction in sinusitis, common cold, otitis media, laryngotracheitis, pharyngitis, stomatitis, and bronchiolitis (Table 2). It is worth mentioning the 87.6% (<0.001) increase in flu-like syndrome in children between the ages of 6 and 12, in contrast to the other diagnoses, which decreased significantly. Acute tonsillitis is the group with the biggest decrease, with 63.9% (*p* < 0.001) (Table 3). In the older age group (12–15 years old), we can see that they present a greater reduction in the incidence of streptococcal pharyngitis than other groups, with 76% (*p* < 0.001), as well as for pneumonia, with 63.8% (*p* < 0.001), and bronchitis, with 61.6% (*p* < 0.001) (Table 3). 

In the diagnoses in rural and urban centres (Table 4), similar prevalences were observed during the 3 years of the study. In 2018 and 2019, the prevalence of flu-like syndrome was similar in both settings, but in 2020 it was higher in the urban setting, with 4.7 (CI: 4.43;4.98), while in the rural environment it was 3.3 (CI: 3.2;3.56). The prevalence of COVID-19 was also higher in urban areas, with 4.2 (CI: 4.03;4.55), while in rural areas it was 3.1 (CI: 2.99;3.34). Diagnoses of acute tonsillitis, bronchitis, pharyngitis, otitis media, common cold, and acute sinusitis in rural centres were higher than in urban centres in all three study years. During the 3 years of the study, otitis media showed a higher prevalence in rural than in urban areas, with a prevalence of 5.2 (CI: 4.98;5.43) and 3.74 (CI: 3.5;3.99), respectively, in 2020. In 2020, COVID had a higher prevalence in urban centres than in rural centres.

In both rural and urban centres, the variation in prevalence between 2019 and 2020 was significant for all respiratory infections, except sinusitis in urban settings. The increase in flu-like syndrome in urban areas was a noteworthy 45.4% (<0.001), compared to 34.2% (<0.001) in rural areas.

## 4. Discussion

The COVID-19 pandemic changed habits and limited social life in an attempt to control contagion. In primary care centres in Central Catalonia, during the first months of the SARS-CoV-2 pandemic, there was a significant decrease in visits to the centres and in the prevalence of respiratory infections. The number of consultations for respiratory infections in the paediatric population decreased by 44.3% compared to 2019, as in France, where Launay et al. [18] observed that consultations for respiratory infections in primary care centres decreased by 49%.

In our study we identified a decrease in all upper air tract infections studied (common cold, sinusitis, pharyngitis, stomatitis, tonsillitis, and otitis).

The prevalence of the common cold decreased by 39.8% with respect to the previous year. The Literature reports that, in Jordan, there was an estimated 21.9% decrease in common colds [19]. In relation to acute tonsillitis, there was a decrease of 60.1% in the first year of the pandemic compared to the previous year. Yamamoto et al. (Japan) reported that tonsillitis cases in children under the age of 15 decreased by 29.3% in this period [20]. We also found a decrease of 54% in the prevalence of acute otitis media, similar to a multi-centre study in Italy, in which Iannella et al. [21] concluded that otitis media decreased by 68.1% between March 2020 and March 2021 with respect to the same period of the previous year. In the Netherlands, primary care diagnoses of acute otitis media decreased by 63% between March 2020 and February 2021, compared to the same period the previous year [22].

In reference to the prevalence of flu-like syndrome in 2020, we observed an increase of 41.7%, which is far from that described in the literature. In China, the Centre for Disease Control and Prevention determined that the seasonal influenza rate in 2020 was 65% lower than in 2019 [23]. Yu et al. [24] also estimated that, in Wuhan, the average influenza positive rate in 2018 was 14%, compared to 28.7% in 2019, with a relevant decrease in 2020 of 4.3%. Said evidence is far from the figures collected in our article, which may be due to the fact that they include flu-like symptoms confirmed by influenza swab tests, while in our study many infections were clinically diagnosed by suggestive symptoms within a compatible epidemic period. In the detailed analysis by month, most of the 2020 flu cases were centred around the months of January and February. As the symptoms are equivalent to those of the SARS-CoV-2 infection, many of the cases classed as influenza-like illnesses may in fact have already been cases of COVID-19.

Focusing on lower air tract infections, we also observed a reduction in the number of diagnoses of pneumonia, bronchitis, bronchiolitis, and laryngotracheitis in 2020. The prevalence of bronchiolitis in 2020 was 0.4%, which represents a 71.1% decrease compared to 2019. These data are similar to those described by Launay et al. [18], who reported a 79% reduction in bronchiolitis compared to data from previous years. At Oslo University Hospital, they reported a significant 90% reduction compared to 2017–2019 [25]. We observed a 57.6% (<0.001) decrease in the prevalence of pneumonia in the first year of the pandemic compared to 2019, which is higher than that described in Thailand, where they reported a 28% decrease in incidents [26].

In our analysis, we obtained a 3.8% (CI: 3.73;4.03) prevalence of COVID-19 in 2020 in those under 15 years of age. In the study of age subgroups, we can see that in children under the age of one, the prevalence was 3.3, similar to pre-schoolers, who had a prevalence of 2.5. The prevalence increases to 3.3 in school children, and the highest prevalence of 4 is detected among adolescents. In Canada, the estimated incidence of COVID-19 in children in 2020 was 15.5% [27]. In a systematic review, Mehta et al. [28] reported that of all COVID cases in China, 0.9% were in children of 0–9 years of age and 1.2% in those of 10–19 years of age. In the Parma University Hospital study, they reported 2% SARS-CoV-2 in the total number of children who consulted for respiratory symptoms between December 2019 and March 2020 [29]. The explanation for these differences could be that these studies were conducted in a hospital setting, while ours was in primary care centres. Another reason could be the multiple protocols that were in place at different evolutionary moments of the pandemic. In the first months of the pandemic, in our study, there were almost no records of cases of COVID-19 in children due to the fact that at that time, in the presence of compatible symptoms, home isolation and symptomatic treatment were carried out without diagnostic testing. As the months progressed, more polymerase chain reaction (PCR) diagnostic tests were performed, followed by rapid antigen tests (RAT), which led to the detection of more cases. Moreover, in younger children, who were at the lowest risk of poor evolution, it was often decided, in agreement with the family, not to perform complementary tests due to the risk/benefit ratio.

An interesting finding in the comparative prevalence of respiratory infections according to age group is the fact that the group with the highest percentage decrease in prevalence for most diagnoses (bronchiolitis, stomatitis, pharyngitis, laryngotracheitis, otitis media, common cold, and acute sinusitis) was the 1- to 6-year-old group. This may be due to the fact that a large part of the prevalence of these respiratory infections is concentrated in this age group since, in the first years of life, the immune system is immature. As exposure to germs was reduced in nurseries, schools, and extracurricular activities, and home isolation was promoted, they present the greatest reduction in prevalence in 2020 with respect to the previous year.

Additionally, the difference between rural and urban centres is noteworthy, with a higher prevalence of acute tonsillitis, bronchiolitis, bronchitis, stomatitis, pharyngitis, laryngotracheitis, otitis media, pneumonia, common cold, and sinusitis was observed in rural primary care centres during the 3 years of the study. In contrast, flu-like syndrome was more prevalent in urban primary care centres between 2018 and 2020. Likewise, in 2020, we can see that COVID-19 was diagnosed more often in the urban environment. No studies have been found to justify these differences between urban and rural centres, but the results could suggest that rural centres have greater accessibility and, consequently, can diagnose more.

### Limitations of the Study

This study has its limitations. As there was no control group, causal relationships cannot be evaluated. However, its observational design allows hypotheses to be formed and adds evidence to the effect of social distancing policies and the use of masks. Another limitation is posed by the difference in diagnostic orientation among the different professionals since these are similar diseases with overlapping clinical symptoms. In many cases, infections are coded for compatible symptomatology without diagnostic tests being performed; this is an important limitation and may be partly related to the marked increase in cases of flu syndrome in 2020, except for the detection of streptococcus and SARS-CoV-2 towards the end of the study period, where diagnostic tests were performed.

As future lines of research, we propose an analysis including the years after the beginning of the pandemic to assess whether COVID-19 displaces other respiratory infections in number or seasonality and to evaluate the trend in primary care visits. Additionally, it would be necessary to carry out a new study to investigate possible causes that justify this decrease and assess whether it is due to home isolation or less accessibility to primary care centres.

## 5. Conclusions

The COVID-19 pandemic caused a decrease in the number of consultations for respiratory infections in the paediatric population of the Central Catalonia region. There was a decrease in the prevalence of all respiratory infections except flu-like syndrome, which increased in January and February 2020. No significant differences in prevalence were found between both sexes, although there were differences in the distribution in the different age groups, with children between the ages of 6 and 12 having the highest prevalence of flu-like syndrome and those between the ages of 12 and 15 presenting the highest prevalence of COVID-19. A higher prevalence of the infections analysed was also observed in rural areas, with the exception of flu-like syndrome and SARS-CoV-2.

## Figures and Tables

**Figure 1 healthcare-11-01252-f001:**
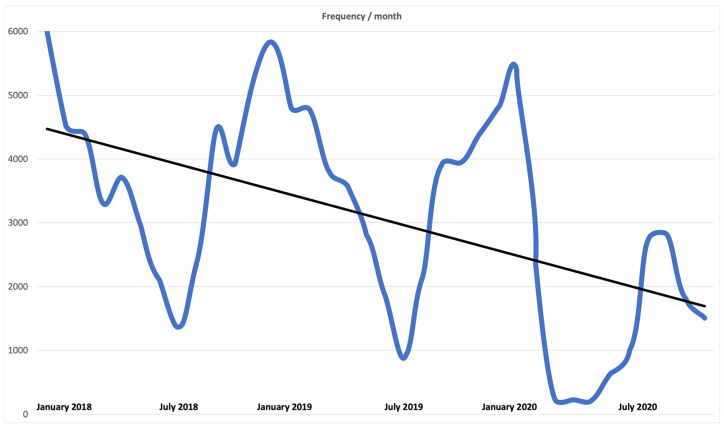
Total number of visits for respiratory infections.

**Table 1 healthcare-11-01252-t001:** Prevalence of respiratory infections.

	2018		2019		2020	
	*n* (%)	CI 95% (%)	n (%)	CI 95% (%)/ *p*-Value	*n* (%)	CI 95% (%)/ *p*-Value
**Acute tonsillitis**	4694 (7.40)	(7.19;7.6)	3972 (6.24)	(6.05;6.43)	1542 (2.47)	(2.35;2.6)
Variation 2019 vs. 2018			−722 (−15.68)	<0.001		
Variation 2020 vs. 2019					−2430 (−60.42)	<0.001
**Streptococcal pharyngitis**	1895 (2.99)	(2.86;3.12)	2818 (4.43)	(4.27;4.59)	831 (1.33)	(1.24;1.43)
Variation 2019 vs. 2018			+923 (+48.16)	<0.001		
Variation 2020 vs. 2019					−1987 (−69.98)	<0.001
**Bronchiolitis**	935 (1.47)	(1.38;1.57)	1041 (1.63)	(1.54;1.74)	296 (0.47)	(0.42;0.53)
Variation 2019 vs. 2018			+106 (+10.88)	0.021		
Variation 2020 vs. 2019					−745 (−71.17)	<0.001
**Bronchitis**	4037 (6.36)	(6.17;6.55)	3647 (5.73)	(5.55;5.91)	1517 (2.43)	(2.31;2.56)
Variation 2019 vs. 2018			−390 (−9.91)	<0.001		
Variation 2020 vs. 2019					−2130 (−57.59)	<0.001
**COVID**	-	-	-	-	2236 (3.58)	(3.44;3.73)
**Stomatitis**	783 (1.23)	(1.15;1.32)	562 (0.88)	(0.81;0.96)	103 (0.17)	(0.14;0.2)
Variation 2019 vs. 2018			−221 (−28.46)	<0.001		
Variation 2020 vs. 2019					−459 (−80.68)	<0.001
**Pharyngitis**	4516 (7.11)	(6.92;7.32)	4445 (6.98)	(6.78;7.18)	1938 (3.11)	(2.97;3.25)
Variation 2019 vs. 2018			−71 (−1.83)	0.353		
Variation 2020 vs. 2019					−2507 (−55.44)	<0.001
**Flu-like syndrome**	1792 (2.82)	(2.7;2.96)	1737 (2.73)	(2.6;2.86)	2417 (3.87)	(3.73;4.03)
Variation 2019 vs. 2018			−55 (−3.19)	0.307		
Variation 2020 vs. 2019					+680 (+41.76)	<0.001
**Laryngotracheitis**	2233 (3.52)	(3.38;3.67)	2782 (4.37)	(4.21;4.53)	1121 (1.8)	(1.7;1.91)
Variation 2019 vs. 2018			549 (24.15)	<0.001		
Variation 2020 vs. 2019					−1661 (−58.81)	<0.001
**Otitis media**	6704 (10.56)	(10.32;10.8)	6441 (10.11)	(9.88;10.35)	2900 (4.65)	(4.49;4.82)
Variation 2019 vs. 2018			−263 (−4.26)	0.009		
Variation 2020 vs. 2019					−3541 (−54.01)	<0.001
**Pneumonia**	689 (1.09)	(1.01;1.17)	704 (1.11)	(1.03;1.19)	295 (0.47)	(0.42;0.53)
Variation 2019 vs. 2018			+15 (+1.83)	0.752		
Variation 2020 vs. 2019					−409 (−57.66)	<0.001
**Common cold**	15,794 (24.88)	(24.55;25.22)	14,290 (22.44)	(22.12;22.77)	8425 (13.51)	(13.24;13.78)
Variation 2019 vs. 2018			−1504 (−9.81)	<0.001		
Variation 2020 vs. 2019					−5865 (−39.8)	<0.001
**Actue sinusitis**	394 (0.62)	(0.56;0.69)	365 (0.57)	(0.52;0.64)	228 (0.37)	(0.32;0.42)
Variation 2019 vs. 2018			−29 (−8.06)	0.287		
Variation 2020 vs. 2019					−137 (−35.09)	<0.001

The change is shown as absolute change and percentage change. The percentage has been calculated as (final value − initial value)/initial value. The *p*-value of the variation has been obtained through the contrast of proportions of the two years compared.

**Table 2 healthcare-11-01252-t002:** Comparison of prevalence of respiratory infections by age group: < 1 years old and 1-6 years old.

	<1 Years Old						1–6 Years Old					
	2018		2019		2020		2018		2019		2020	
	*n* (%)	CI 95% (%)	*n* (%)	CI 95% (%)/ *p*-Value	*n* (%)	CI 95% (%)/ *p*-Value	*n* (%)	CI 95% (%)	*n* (%)	CI 95% (%)/ *p*-Value	*n* (%)	CI 95% (%)/ *p*-Value
**Acute tonsillitis**	36 (1.15)	(0.82;1.61)	32 (1.05)	(0.73;1.49)	12 (0.5)	(0.27;0.9)	1732 (8.97)	(8.58;9.39)	1411 (7.3)	(6.94;7.67)	595 (3.16)	(2.92;3.42)
Variation 2019 vs. 2018			−4 (−8.7)	0.779					−321 (−18.62)	<0.001		
Variation 2020 vs. 2019					−20 (−52.38)	0.037					−816 (−56.71)	<0.001
**Streptococcal pharyngitis**	4 (0.13)	(0.04;0.35)	1 (0.03)	(0;0.21)	1 (0.04)	(0;0.27)	517 (2.68)	(2.46;2.92)	769 (3.98)	(3.71;4.26)	264 (1.4)	(1.24;1.58)
Variation 2019 vs. 2018			−3 (−76.92)	0.383					252 (48.51)	<0.001		
Variation 2020 vs. 2019					0 (33.33)	1					−505 (−64.82)	<0.001
**Bronchiolitis**	664 (21.25)	(19.84;22.74)	803 (26.24)	(24.7;27.85)	219 (9.14)	(8.03;10.38)	262 (1.36)	(1.2;1.53)	231 (1.19)	(1.05;1.36)	72 (0.38)	(0.3;0.48)
Variation 2019 vs. 2018			139 (23.48)	<0.001					−31 (−12.5)	0.167		
Variation 2020 vs. 2019					−584 (−65.17)	<0.001					−159 (−68.07)	<0.001
**Bronchitis**	417 (13.35)	(12.19;14.6)	360 (11.76)	(10.66;12.97)	148 (6.18)	(5.26;7.24)	2619 (13.57)	(13.09;14.06)	2375 (12.28)	(11.82;12.75)	921 (4.89)	(4.59;5.22)
Variation 2019 vs. 2018			−57 (−11.91)	0.066					−244 (−9.51)	<0.001		
Variation 2020 vs. 2019					−212 (−47.45)	<0.001					−1454 (−60.18)	<0.001
**COVID**	-	-	-	-	80 (3.34)	(2.67;4.16)	-	-	-	-	480 (2.55)	(2.33;2.79)
**Stomatitis**	80 (2.56)	(2.05;3.19)	49 (1.6)	(1.2;2.13)	14 (0.58)	(0.33;1)	673 (3.49)	(3.23;3.76)	489 (2.53)	(2.31;2.76)	82 (0.44)	(0.35;0.54)
Variation 2019 vs. 2018			−31 (−37.5)	0.011					−184 (−27.51)	<0.001		
Variation 2020 vs. 2019					−35 (−63.75)	0.001					−407 (−82.61)	<0.001
**Pharyngitis**	167 (5.35)	(4.6;6.21)	139 (4.54)	(3.84;5.36)	70 (2.92)	(2.3;3.7)	1626 (8.42)	(8.04;8.83)	1609 (8.32)	(7.94;8.72)	567 (3.01)	(2.78;3.27)
Variation 2019 vs. 2018			−28 (−15.14)	0.162					−17 (−1.19)	0.726		
Variation 2020 vs. 2019					−69 (−35.68)	0.002					−1042 (−63.82)	<0.001
**Flu-like syndromes**	54 (1.73)	(1.31;2.27)	57 (1.86)	(1.43;2.42)	42 (1.75)	(1.28;2.38)	788 (4.08)	(3.81;4.37)	824 (4.26)	(3.98;4.56)	958 (5.09)	(4.78;5.42)
Variation 2019 vs. 2018			3 (7.51)	0.763					36 (4.41)	0.395		
Variation 2020 vs. 2019					−15 (−5.91)	0.842					134 (19.48)	<0.001
**Laryngotracheitis**	243 (7.78)	(6.88;8.79)	332 (10.85)	(9.78;12.02)	112 (4.67)	(3.88;5.62)	1354 (7.02)	(6.66;7.39)	1708 (8.83)	(8.44;9.24)	605 (3.22)	(2.97;3.48)
Variation 2019 vs. 2018			89 (39.46)	<0.001					354 (25.78)	<0.001		
Variation 2020 vs. 2019					−220 (−56.96)	<0.001					−1103 (−63.53)	<0.001
**Otitis media**	514 (16.45)	(15.18;17.81)	473 (15.46)	(14.2;16.8)	230 (9.6)	(8.46;10.87)	4186 (21.69)	(21.11;22.28)	4012 (20.75)	(20.18;21.33)	1697 (9.02)	(8.62;9.44)
Variation 2019 vs. 2018			−41 (−6.02)	0.301					−174 (−4.33)	0.024		
Variation 2020 vs. 2019					−243 (−37.9)	<0.001					−2315 (−56.53)	<0.001
**Pneumonia**	29 (0.93)	(0.63;1.35)	31 (1.01)	(0.7;1.45)	17 (0.71)	(0.43;1.16)	416 (2.16)	(1.96;2.37)	431 (2.23)	(2.03;2.45)	172 (0.91)	(0.79;1.06)
Variation 2019 vs. 2018			2 (8.6)	0.833					15 (3.24)	0.647		
Variation 2020 vs. 2019					−14 (−29.7)	0.296					−259 (−59.19)	<0.001
**Common cold**	2861 (91.58)	(90.54;92.52)	2565 (83.82)	(82.46;85.1)	1076 (44.91)	(42.91;46.93)	7981 (41.35)	(40.66;42.05)	7304 (37.77)	(37.09;38.46)	3590 (19.08)	(18.52;19.65)
Variation 2019 vs. 2018			−296 (−8.47)	<0.001					−677 (−8.66)	<0.001		
Variation 2020 vs. 2019					−1489 (−46.42)	<0.001					−3714 (−49.48)	<0.001
**Actue sinusitis ***	1 (0.03)	(0;0.21)	-	-	-	-	38 (0.2)	(0.14;0.27)	23 (0.12)	(0.08;0.18)	6 (0.03)	(0.01;0.07)
Variation 2019 vs. 2018									−15 (−40)	0.072		
Variation 2020 vs. 2019											−17 (−75)	<0.001

The change is shown as absolute change and percentage change. The percentage has been calculated as (final value − initial value)/initial value. The *p*-value of the variation has been obtained through the contrast of proportions of the two compared years. * No cases of acute sinusitis in children < 1 year old in 2019 and 2020.

**Table 3 healthcare-11-01252-t003:** Comparison of prevalence of respiratory infections by age group: 6-12 years old and 12-15 years old.

	6–12 Years Old						12–15 Years Old					
	2018		2019		2020		2018		2019		2020	
	*n* (%)	CI 95% (%)	*n* (%)	CI 95% (%)/ *p*-Value	*n* (%)	CI 95% (%)/ *p*-Value	*n* (%)	CI 95% (%)	*n* (%)	CI 95% (%)/ *p*-Value	*n* (%)	CI 95% (%)/ *p*-Value
**Acute tonsillitis**	2151 (7.83)	(7.52;8.16)	1825 (6.66)	(6.37;6.97)	647 (2.4)	(2.23;2.6)	775 (4.32)	(4.03;4.63)	704 (3.81)	(3.54;4.1)	288 (1.52)	(1.36;1.71)
Variation 2019 vs. 2018			−326 (−14.94)	<0.001					−71 (−11.81)	0.016		
Variation 2020 vs. 2019					−1178 (−63.96)	<0.001					−416 (−60.1)	<0.001
**Streptococcal pharyngitis**	1059 (3.86)	(3.63;4.09)	1577 (5.76)	(5.49;6.04)	451 (1.68)	(1.53;1.84)	315 (1.75)	(1.57;1.96)	471 (2.55)	(2.33;2.79)	115 (0.61)	(0.5;0.73)
Variation 2019 vs. 2018			518 (49.22)	<0.001					156 (45.71)	<0.001		
Variation 2020 vs. 2019					−1126 (−70.83)	<0.001					−356 (−76.08)	<0.001
**Bronchiolitis**	6 (0.02)	(0.01;0.05)	6 (0.02)	(0.01;0.05)	3 (0.01)	(0;0.04)	3 (0.02)	(0;0.05)	1 (0.01)	(0;0.04)	2 (0.01)	(0;0.04)
Variation 2019 vs. 2018			0 (0)	1					−2 (−50)	0.597		
Variation 2020 vs. 2019					−3 (−50)	0.522					1 (0)	1
**Bronchitis**	780 (2.84)	(2.65;3.04)	706 (2.58)	(2.39;2.77)	366 (1.36)	(1.23;1.51)	221 (1.23)	(1.08;1.41)	206 (1.12)	(0.97;1.28)	82 (0.43)	(0.35;0.54)
Variation 2019 vs. 2018			−74 (−9.15)	0.062					−15 (−8.94)	0.329		
Variation 2020 vs. 2019					−340 (−47.29)	<0.001					−124 (−61.61)	<0.001
**COVID**	-	-	-	-	908 (3.37)	(3.16;3.6)	-	-	-	-	768 (4.06)	(3.79;4.35)
**Stomatitis ****	30 (0.11)	(0.08;0.16)	23 (0.08)	(0.05;0.13)	6 (0.02)	(0.01;0.05)	-	-	1 (0.01)	(0;0.04)	1 (0.01)	(0;0.03)
Variation 2019 vs. 2018			−7 (−27.27)	0.415					-	-		
Variation 2020 vs. 2019					−17 (−75)	0.003					-	-
**Pharyngitis**	1865 (6.79)	(6.5;7.1)	1823 (6.65)	(6.36;6.96)	847 (3.15)	(2.94;3.37)	858 (4.78)	(4.47;5.1)	874 (4.73)	(4.43;5.05)	454 (2.4)	(2.19;2.63)
Variation 2019 vs. 2018			−42 (−2.06)	0.537					16 (−1.05)	0.853		
Variation 2020 vs. 2019					−976 (−52.63)	<0.001					−420 (−49.26)	<0.001
**Flu-like syndromes**	629 (2.29)	(2.12;2.48)	557 (2.03)	(1.87;2.21)	1024 (3.81)	(3.58;4.04)	321 (1.79)	(1.6;1.99)	299 (1.62)	(1.44;1.81)	393 (2.08)	(1.88;2.29)
Variation 2019 vs. 2018			−72 (−11.35)	0.041					−22 (−9.5)	0.228		
Variation 2020 vs. 2019					467 (87.68)	<0.001					94 (28.4)	0.001
**Laryngotracheitis**	534 (1.94)	(1.79;2.12)	591 (2.16)	(1.99;2.34)	334 (1.24)	(1.11;1.38)	102 (0.57)	(0.47;0.69)	151 (0.82)	(0.69;0.96)	70 (0.37)	(0.29;0.47)
Variation 2019 vs. 2018			57 (11.34)	0.083					49 (43.86)	0.005		
Variation 2020 vs. 2019					−257 (−42.59)	<0.001					−81 (−54.88)	<0.001
**Otitis media**	1561 (5.68)	(5.41;5.97)	1499 (5.47)	(5.21;5.75)	706 (2.62)	(2.44;2.82)	443 (2.47)	(2.25;2.71)	457 (2.47)	(2.26;2.71)	267 (1.41)	(1.25;1.59)
Variation 2019 vs. 2018			−62 (−3.7)	0.289					14 (0)	0.993		
Variation 2020 vs. 2019					−793 (−52.1)	<0.001					−190 (−42.91)	<0.001
**Pneumonia**	190 (0.69)	(0.6;0.8)	176 (0.64)	(0.55;0.75)	82 (0.3)	(0.24;0.38)	54 (0.3)	(0.23;0.4)	66 (0.36)	(0.28;0.46)	24 (0.13)	(0.08;0.19)
Variation 2019 vs. 2018			−14 (−7.25)	0.511					12 (20)	0.395		
Variation 2020 vs. 2019					−94 (−53.13)	<0.001					−42 (−63.89)	<0.001
**Common cold**	3534 (12.87)	(12.47;13.27)	3183 (11.62)	(11.24;12.01)	2677 (9.95)	(9.59;10.31)	1418 (7.9)	(7.51;8.3)	1238 (6.7)	(6.35;7.07)	1082 (5.72)	(5.4;6.06)
Variation 2019 vs. 2018			−351 (−9.71)	<0.001					−180 (−15.19)	<0.001		
Variation 2020 vs. 2019					−506 (−14.37)	<0.001					−156 (−14.63)	<0.001
**Actue sinusitis**	178 (0.65)	(0.56;0.75)	152 (0.55)	(0.47;0.65)	114 (0.42)	(0.35;0.51)	177 (0.99)	(0.85;1.14)	190 (1.03)	(0.89;1.19)	108 (0.57)	(0.47;0.69)
Variation 2019 vs. 2018			−26 (−15.38)	0.175					13 (4.04)	0.721		
Variation 2020 vs. 2019					−38 (−23.64)	0.033					−82 (−44.66)	<0.001

The change is shown as absolute change and percentage change. The percentage has been calculated as (final value − initial value)/initial value. The *p*-value of the variation has been obtained through the contrast of proportions of the two compared years. ** No cases of stomatitis in children aged 12–15 in 2018.

**Table 4 healthcare-11-01252-t004:** Prevalence of respiratory infections according to rural and urban centres.

	Rural						Urban					
	2018		2019		2020		2018		2019		2020	
	*n* (%)	CI 95% (%)	*n* (%)	CI 95% (%)/ *p*-Value	*n* (%)	CI 95% (%)/ *p*-Value	*n* (%)	CI 95% (%)	*n* (%)	CI 95% (%)/ *p*-Value	*n* (%)	CI 95% (%)/ *p*-Value
**Acute tonsillitis**	3414 (8.58)	(8.31;8.86)	2926 (7.37)	(7.12;7.64)	1071 (2.76)	(2.6;2.93)	1280 (5.4)	(5.12;5.7)	1046 (4.36)	(4.11;4.63)	471 (2)	(1.82;2.19)
Variation 2019 vs. 2018			−488 (−14.29)	<0.001					−234 (−18.28)	<0.001		
Variation 2020 vs. 2019					−1855 (−63.4)	<0.001					−575 (−54.97)	<0.001
**Streptococcal pharyngitis**	1222 (3.07)	(2.91;3.25)	1943 (4.9)	(4.69;5.11)	592 (1.53)	(1.41;1.65)	673 (2.84)	(2.63;3.06)	875 (3.65)	(3.41;3.89)	239 (1.01)	(0.89;1.15)
Variation 2019 vs. 2018			721 (59)	<0.001					202 (30.01)	<0.001		
Variation 2020 vs. 2019					−1351 (−69.53)	<0.001					−636 (−72.69)	<0.001
**Bronchiolitis**	590 (1.48)	(1.37;1.61)	674 (1.7)	(1.57;1.83)	198 (0.51)	(0.44;0.59)	345 (1.46)	(1.31;1.62)	367 (1.53)	(1.38;1.69)	98 (0.42)	(0.34;0.51)
Variation 2019 vs. 2018			84 (14.24)	0.017					22 (6.38)	0.535		
Variation 2020 vs. 2019					−476 (−70.62)	<0.001					−269 (−73.3)	<0.001
**Bronchitis**	2736 (6.88)	(6.63;7.13)	2682 (6.76)	(6.51;7.01)	1086 (2.8)	(2.64;2.97)	1301 (5.49)	(5.21;5.79)	965 (4.02)	(3.78;4.28)	431 (1.83)	(1.66;2.01)
Variation 2019 vs. 2018			−54 (−1.97)	0.514					−336 (−25.83)	<0.001		
Variation 2020 vs. 2019					−1596 (−59.51)	<0.001					−534 (−55.34)	<0.001
**COVID**	-	-	-	-	1226 (3.16)	(2.99;3.34)	-	-	-	-	1010 (4.28)	(4.03;4.55)
**Stomatitis**	555 (1.4)	(1.28;1.52)	347 (0.87)	(0.79;0.97)	72 (0.19)	(0.15;0.24)	228 (0.96)	(0.84;1.1)	215 (0.9)	(0.78;1.03)	31 (0.13)	(0.09;0.19)
Variation 2019 vs. 2018			−208 (−37.48)	<0.001					−13 (−5.7)	0.478		
Variation 2020 vs. 2019					−275 (−79.25)	<0.001					−184 (−85.58)	<0.001
**Pharyngitis**	3209 (8.07)	(7.8;8.34)	3004 (7.57)	(7.31;7.84)	1349 (3.48)	(3.3;3.66)	1307 (5.52)	(5.23;5.82)	1441 (6)	(5.71;6.31)	589 (2.5)	(2.3;2.71)
Variation 2019 vs. 2018			−205 (−6.39)	0.009					134 (10.25)	0.023		
Variation 2020 vs. 2019					−1655 (−55.09)	<0.001					−852 (−59.13)	<0.001
**Flu-like syndromes**	1109 (2.79)	(2.63;2.96)	975 (2.46)	(2.31;2.62)	1309 (3.37)	(3.2;3.56)	683 (2.88)	(2.68;3.11)	762 (3.17)	(2.96;3.41)	1108 (4.7)	(4.43;4.98)
Variation 2019 vs. 2018			−134 (−12.08)	0.004					79 (11.57)	0.066		
Variation 2020 vs. 2019					334 (34.26)	<0.001					346 (45.41)	<0.001
**Laryngotracheitis**	1500 (3.77)	(3.59;3.96)	1843 (4.64)	(4.44;4.86)	707 (1.82)	(1.69;1.96)	733 (3.09)	(2.88;3.32)	939 (3.91)	(3.67;4.17)	414 (1.76)	(1.59;1.93)
Variation 2019 vs. 2018			343 (22.87)	<0.001					206 (28.1)	<0.001		
Variation 2020 vs. 2019					−1136 (−61.64)	<0.001					−525 (−55.91)	<0.001
**Otitis media**	4557 (11.46)	(11.15;11.77)	4354 (10.97)	(10.67;11.29)	2018 (5.2)	(4.98;5.43)	2147 (9.06)	(8.7;9.44)	2087 (8.7)	(8.34;9.06)	882 (3.74)	(3.5;3.99)
Variation 2019 vs. 2018			−203 (−4.45)	0.032					−60 (−2.79)	0.165		
Variation 2020 vs. 2019					−2336 (−53.65)	<0.001					−1205 (−57.74)	<0.001
**Pneumonia**	454 (1.14)	(1.04;1.25)	482 (1.21)	(1.11;1.33)	192 (0.49)	(0.43;0.57)	235 (0.99)	(0.87;1.13)	222 (0.92)	(0.81;1.06)	103 (0.44)	(0.36;0.53)
Variation 2019 vs. 2018			28 (6.17)	0.355					−13 (−5.53)	0.482		
Variation 2020 vs. 2019					−290 (−60.17)	<0.001					−119 (−53.6)	<0.001
**Common cold**	10,154 (25.53)	(25.1;25.96)	9281 (23.39)	(22.97;23.81)	5808 (14.97)	(14.62;15.33)	5640 (23.8)	(23.26;24.35)	5009 (20.87)	(20.36;21.39)	2617 (11.1)	(10.7;11.51)
Variation 2019 vs. 2018			−873 (−8.6)	<0.001					−631 (−11.19)	<0.001		
Variation 2020 vs. 2019					−3473 (−37.42)	<0.001					−2392 (−47.75)	<0.001
**Actue sinusitis**	295 (0.74)	(0.66;0.83)	271 (0.68)	(0.61;0.77)	164 (0.42)	(0.36;0.49)	99 (0.42)	(0.34;0.51)	94 (0.39)	(0.32;0.48)	64 (0.27)	(0.21;0.35)
Variation 2019 vs. 2018			−24 (−8.14)	0.347					−5 (−5.05)	0.705		
Variation 2020 vs. 2019					−107 (−39.48)	<0.001					−30 (−31.91)	0.028

The change is shown as absolute change and percentage change. The percentage has been calculated as (final value − initial value)/initial value. The *p*-value of the variation has been obtained through the contrast of proportions of the two compared years.

## Data Availability

The data that support the findings of this study are available from the corresponding author upon request.

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
