# Peer review of "Prevalence of Respiratory Infections during the 2018–2020 Period in the Paediatric Population of Primary Care Centres in Central Catalonia"

_healthcare, 2023, doi:10.3390/healthcare11091252_

Round 1
Reviewer 1 Report
Major Comments:
1. L. 278-279: Is there any SARS-CoV-2 circulation in Catalonia during January-February 2020? Were there restrictive measures implemented so that they may have affected the circulation of respiratory pathogens? To be clarified.
The following should be reported:
- When has SARS-CoV-2 appeared in Catalonia?
- Which part of 2020 do the respiratory cases belong to? Does the majority of them co-exist with COVID-19 or does it precede the SARS-CoV-2 circulation in the area? To be clarified, otherwise the total of findings cannot be evaluated!
Minor comments:
1. L. 20: Of course, you mean 44.35%. To be added.
2. The report will be fully documented if the numbers of cases are presented, not only percentages. To be added.
3. L. 165: The label should mention the periods (years) that are compared to each other.
4. L. 171-172: The prevalence of COVID-19 in urban and rural areas is vice versa: urban 4.28, rural 3.16. To be corrected.
Author Response
"Please see the attachment."

Reviewer 2 Report
The article provides a compilation of some epidemiological data that, despite the obvious conclusions, confirm assumptions and observations in other countries.
The fundamental question - whether the observed declines in doctor visits were due to a lack of need for medical consultations or to impaired access to health care - was not addressed.
What was the availability of doctors, whether visits were standard or telephone consultations, if so, at what percentage. Did the reduced availability of health care translate into reductions in diagnoses? Did the decrease in visits due to obstruction result in a reduction in cases diagnosed by remaining patients at home?
In the discussion, the authors propose an analysis covering further years, but it would be worthwhile to try more to explain the observed phenomena under the influence of changes or new factors created by the pandemic, e.g. the authors did not address the reasons for the decrease in the frequency of most respiratory infections and the increase in the frequency of influenza infections. A similar phenomenon is also observed in the case of other infections (including bacterial), the frequency of which is increasing.
Abstract:
- "In 19 2020, there was a 44.35 decrease in total consultations for respiratory infections compared to 2019." What does 44.35 decrease mean? What is the parameter?
- "All respiratory infections exhibited a significant decrease except influenza, which increased by 41.76%." All respiratory infections showed a decrease or incidence?
- "In conclusion, the COVID-19 pandemic caused a significant decrease in the number of consultations for respiratory infections in the paediatric population, except for influenza, which increased in cases in January and February 2020. No differences were found between sexes, although differences were found in the distribution of the different age groups." Unintelligible, illogical sentence. What is the relationship between the decrease in the number of consultations for respiratory infections in the population except influenza and the increase in influenza? That is, the frequency of consultations regarding suspected influenza has increased ? What does it mean that influenza is increasing ? Does it mean the frequency of diagnosed cases ?
Don't the authors distinguish between the frequency of visits and the frequency of diagnosed cases?
The type of diagnosis also matters. What matters is whether the majority of cases were diagnosed symptomatically or whether the potential diagnosis was confirmed by a laboratory test, and then only then can we talk about confirmed cases of diagnosis.
Materials and Methods
What statistical tests were used? What testing algorithm was adopted?
- Line 110 - it is not accepted to mark a decrease or increase in % with "+" and "-" signs if we determine whether it is an increase or decrease. In the text it is good to give % values rounded to integers, e.g. about, almost, close to, more than 44%. It is better to remember such values, the accuracy after does not matter and in the conclusions it does not change anything if it is not data for calculations. If accuracy is important in the considerations then to one decimal place at most.
- Figure 1 - is of poor quality. Little readable is the X axis. Please try to add trend lines to the graph.
-lines 122-123 - what measure of central tendency do the authors give? Is it the arithmetic mean ? If so, what is the justification for its use ? Why is this information not in Materials and methods ?
-Table 1 - The table is poorly constructed. Why does the text of the table headings appear in vertical orientation ? What does "Variation 18-19" and "Variation 19-21" mean ? ? If these are years then this is a bad designation. In the table, providing + and - signs is appropriate.
How was the variation parameter calculated and what does it mean ? The results posted do not correspond to the difference in the frequency of diagnosed cases given in the columns for the corresponding years.
What test was used to calculate the p-value. You can provide the information in small print below the table.
Please consider rebuilding the table by adding 2 more rows under each diagnosis and writing the % change under the corresponding year and p-value between the corresponding years.
Why is the text of diagnoses in the 1st column written in capital letters ? The data in the table has become disjointed.
Why is the value for :Streptococcal" missing ?
Notes to the Tables in the supplementary are the same.
- Line 132 - please state what period the changes refer to. No information on how they were calculated.
- Line 138- Perhaps it would be better to state that no correlation was observed between the incidence of infections and the age of the children analyzed?
- Line 139-149- What are the SA units of variation and how were they calculated ?
-Line 150-162 - Table 2 does not state as written by the authors
- Table 2 - The title of the table "Comparison of prevalence of respiratory infections by age group" does not correlate with its content. Only the variability is given in the table. How was the variation parameter calculated and what does it mean? With what test was the p-value calculated. Why is the text of diagnoses in the 1st column written in capital letters ? Why is the value for :Streptococcal" missing ?
-Table 3 - How was the change parameter calculated and what does it mean? What test was used to calculate the p-value. Why is the text of diagnoses in the 1st column written in capital letters ? Why is the value for :Streptococcal" missing ? Please consider rebuilding the table by adding 2 more rows under each diagnosis and writing % of change under the corresponding year and p-value between the corresponding years.
Please separate and clarify the point about the limitations of the study in the results obtained and on this basis try to assess the reliability of the data and conclusions obtained
Author Response
"Please see the attachment."

Round 2
Reviewer 1 Report
Minor comments:
1. The word “influenza” should be changed to “flu-like syndrome” or “ILI” not only in the text but on the tables too.
2. Table 3: The increase of “Influenza” is presented to be 45.41% and 34.26% in urban and rural populations respectively, instead of 48.2% and 36.9% written in the paragraph below the table. Which are the correct numbers? To be corrected.
Author Response
Thank you for your comments. We have modified the manuscript accordingly